# Serum High-Mobility Group Box 1 and Heme Oxygenase-1 as Biomarkers in COVID-19 Patients at Hospital Admission

**DOI:** 10.3390/ijms241713164

**Published:** 2023-08-24

**Authors:** Ilijana Grigorov, Snežana Pejić, Ana Todorović, Dunja Drakulić, Filip Veljković, Jadranka Miletić Vukajlović, Katarina Bobić, Ivan Soldatović, Siniša Đurašević, Nebojša Jasnić, Sanja Stanković, Sofija Glumac, Violeta Mihailović-Vučinić, Branislava Milenković

**Affiliations:** 1Institute for Biological Research “Siniša Stanković”—National Institute of the Republic of Serbia, University of Belgrade, 11000 Belgrade, Serbia; 2Vinča Institute of Nuclear Sciences—National Institute of the Republic of Serbia, University of Belgrade, 11000 Belgrade, Serbia; snezana@vin.bg.ac.rs (S.P.); anato@vin.bg.ac.rs (A.T.); drakulic@vin.bg.ac.rs (D.D.); filipveljkovic@vin.bg.ac.rs (F.V.); jadranka@vin.bg.ac.rs (J.M.V.); katarina.bobic@vin.bg.ac.rs (K.B.); 3Institute of Medical Statistics and Informatic, Faculty of Medicine, University of Belgrade, 11000 Belgrade, Serbia; ivan.soldatovic@med.bg.ac.rs; 4Faculty of Biology, University of Belgrade, 11000 Belgrade, Serbia; sine@bio.bg.ac.rs (S.Đ.); jasnicn@bio.bg.ac.rs (N.J.); 5Center for Medical Biochemistry, University Clinical Center of Serbia, 11000 Belgrade, Serbia; sanjast2013@gmail.com; 6Institute of Pathology, School of Medicine, University of Belgrade, 11000 Belgrade, Serbia; sofijaglumac09@gmail.com (S.G.); violeta.vucinic@kcs.ac.rs (V.M.-V.); branislava.milenkovic@kcs.ac.rs (B.M.); 7Clinic for Pulmonary Diseases, University Clinical Center of Serbia, Faculty of Medicine, University of Belgrade, 11000 Belgrade, Serbia

**Keywords:** SARS-CoV-2, COVID-19, high-mobility group box 1(HMGB1), heme oxygenase-1 (HO-1), predictive biomarker

## Abstract

The careful monitoring of patients with mild/moderate COVID-19 is of particular importance because of the rapid progression of complications associated with COVID-19. For prognostic reasons and for the economic management of health care resources, additional biomarkers need to be identified, and their monitoring can conceivably be performed in the early stages of the disease. In this retrospective cross-sectional study, we found that serum concentrations of high-mobility group box 1 (HMGB1) and heme oxygenase-1 (HO-1), at the time of hospital admission, could be useful biomarkers for COVID-19 management. The study included 160 randomly selected recovered patients with mild to moderate COVID-19 on admission. Compared with healthy controls, serum HMGB1 and HO-1 levels increased by 487.6 pg/mL versus 43.1 pg/mL and 1497.7 pg/mL versus 756.1 pg/mL, respectively. Serum HO-1 correlated significantly with serum HMGB1, oxidative stress parameters (malondialdehyde (MDA), the phosphatidylcholine/lysophosphatidylcholine ratio (PC/LPC), the ratio of reduced and oxidative glutathione (GSH/GSSG)), and anti-inflammatory acute phase proteins (ferritin, haptoglobin). Increased heme catabolism/hemolysis were not detected. We hypothesize that the increase in HO-1 in the early phase of COVID-19 disease is likely to have a survival benefit by providing protection against oxidative stress and inflammation, whereas the level of HMGB1 increase reflects the activity of the innate immune system and represents levels within which the disease can be kept under control.

## 1. Introduction

Severe Acute Respiratory Syndrome Coronavirus 2 (SARS-CoV-2) develops a robust systemic inflammatory response that can lead to lung injury and multisystem organ dysfunction [1]. The spectrum of disease can range from asymptomatic infection to severe pneumonia with acute respiratory distress syndrome (ARDS) and death. According to the NIH guidelines for coronavirus disease-19 [2], adults with COVID-19 can be classified into five severity categories (asymptomatic, mild, moderate, severe, and critical). Although clinical trials focus on severe and critical COVID-19 patients, those with COVID-19-related lower respiratory/intermediate disease are of particular concern because lung disease can progress rapidly and lead to life-threatening complications. The close monitoring of these individuals has previously included measurements of additional inflammatory markers that may have prognostic value, such as C-reactive protein (CRP), D-dimer, and ferritin. In addition, it is thought that high-mobility group box 1 (HMGB1), a well-known damage-associated molecular pattern protein (DAMP), may be another important biomarker of lung injury and exacerbation of COVID-19-related disease [3,4]. 

HMGB1 is a nonhistone chromatin-associated protein with diverse nuclear, cytoplasmic, and extracellular actions. In addition to its function as a DNA chaperone and cell death regulator in the nucleus and cytoplasm [5], extracellular HMGB1 plays a critical role in extracellular signaling associated with the production of proinflammatory cytokines and the amplification of inflammation [6,7,8]. It is passively released from infected and necrotic nonimmune cells and actively released in the response of immune cells to cytokines (Tumor necrosis factorα (TNFɑ), Interleukin-6 (IL-6), Interferon-γ (IFN-γ)) and the bacterial endotoxin lipopolysaccharide (LPS). An excessive release of HMGB1 from activated immune cells leads to tissue damage [9]. Reactive oxygen species (ROS) are also important mediators in the release of HMGB1 from both immune and nonimmune cells [10]. The release of HMGB1 is also associated with various viral infections [11]. In the era of COVID-19, HMGB1 is considered a possible major player in the SARS-CoV-2-related triggering of hypercytokinemia, cytokine storms, and immunothrombosis, underlying the development of acute lung injury (ALI) and ARDS [12,13].

Elevated serum HMGB1 levels have been reported in hospitalized COVID-19 patients with severe and critical illness [14,15] and in COVID-19 patients suffering from headaches [16]. Chen et al. [17] found in their study that elevated HMGB1 serum levels at hospitalization were correlated with poor clinical outcomes in patients with COVID-19. Despite this study, little is known about serum HMGB1 levels in COVID-19 patients who had moderate symptoms on hospital admission and in whom the disease had not complicated. Therefore, the determination of the serum HMGB1 level in these patients would be important for the possible therapeutic maintenance of the level within limits that would prevent the occurrence of complications.

In addition to the HMGB1 protein, the recent literature indicates that serum levels of heme oxygenase-1 (HO-1) should also be determined in COVID-19 patients because it can prevent the consequences of hemolysis. Hemolysis in patients with COVID-19 occurs as a result of the direct destruction of red blood cells (RBCs) by SARS-CoV-2 [18] and by the formation of autoantibodies against viral proteins [19]. An increase in the red cell distribution width, as a marker of hemolysis [20], and a decrease in serum haptoglobin (Hp) were found in patients with severe COVID-19 [21], as well as a significant increase in HO-1 expression in the plasma of severely ill patients with COVID-19 [22].

The accumulation of hemolytic products, such as hemoglobin and heme, and their delayed scavenging by haptoglobin (Hp) and hemopexin may exacerbate the inflammatory response as a result of the increased production of ROS, tissue damage, and the increased release of DAMPs, such as HMGB1 [23]. Haptoglobin, an acute phase protein, in addition to its primary role in hemoglobin binding, can also inactivate extracellular HMGB1 and block HMGB1-mediated thrombotic and inflammatory responses [24]. The HMGB1-Hp complex activates macrophages to produce and release the anti-inflammatory mediator Interleukin-10 (IL-10), which counteracts the action of excess proinflammatory cytokines in sterile and infectious inflammatory conditions [24]. An important role in neutralizing hemolytic products is played by cellular defense mechanisms that include HO-1, which degrades heme to free iron (leading to an increase in ferritin synthesis), biliverdin, and carbon monoxide (CO). Biliverdin is enzymatically converted to bilirubin, which, similar to CO, has antioxidant potential [25]. Recent studies have shown that HO-1 can reduce viral replication and exerts antiviral activities against a variety of viruses [26]. Given its antiviral, anti-inflammatory, antioxidant, and overall cytoprotective potential, Wagener et al. [27] suggested that inducing the expression of HO-1 could prevent SARS-CoV-2-induced pulmonary complications.

In some hospitalized patients with COVID-19, the severity of laboratory and imaging findings did not correlate with the mild to moderate clinical manifestations they presented with, making them candidates for special surveillance [21]. Such cases highlight the need to identify additional biomarkers for the early stages of the disease, not only for prognostic reasons but also for a more economical use of healthcare resources. Therefore, this study focused primarily on the investigation of serum levels of HMGB1 and HO-1 in COVID-19 patients with mild to moderate clinical symptoms on hospital admission and their correlations with each other and with other biochemical, inflammatory, and oxidative parameters with an attempt to evaluate their potential as biomarkers for COVID-19 management.

## 2. Results

Mean values of serum concentrations of HMGB1 and HO-1 determined in mild/moderate COVID-19 patients on admission to the hospital (Table 1) show that there was already a statistically significant increase in their concentrations in the first days of COVID-19 disease, especially of HMGB1, compared with non-COVID-19 subjects. Because there were no significant differences in HMGB1 and HO-1 levels between patients with mild and moderate COVID-19 (Appendix A), we studied mild to moderate patients as a single group. A gender difference was observed only for HO-1 (Table 2), whose concentration was significantly lower in COVID-19 patients who were women compared with men. The correlation between serum concentrations of HMGB1 and HO-1 in COVID-19 patients (Figure 1) was positive and statistically significant (r = 0.264, *p* < 0.001).

Comparison of patients with COVID-19 with subjects without COVID-19 revealed statistically significant differences in all oxidative stress parameters studied (Table 3). The increased value of prooxidant–antioxidant balance (PAB) in COVID-19 patients indicates a disturbed redox balance and increased prooxidant status, which may lead to the progressive damage of biomolecules such as proteins and lipids. Accordingly, we found increased levels of advanced oxidation protein products (AOPPs), indicating oxidative damage to proteins. A decrease in total antioxidant capacity in patients with COVID-19 is also indicated by a decrease in the ratio of reduced and oxidative glutathione (GSH/GSSG), which tends to decrease under oxidative stress conditions. We also found significantly increased levels of malondialdehyde (MDA) and 4-hydroxynonenal (HNE), the major lipid peroxidation products produced by the peroxidation of unsaturated fatty acids or other lipids.

Changes in blood phospholipid composition indicate the development of inflammation, and the ratio of phosphatidylcholine and lysophosphatidylcholine (PC/LPC) in blood serum is increasingly considered a significant lipid biomarker of inflammation [28]. In particular, LPC derived from PC by the enzymatic action of phospholipase A2 (PLA2) or the release of ROS play an important role in promoting inflammatory responses and subsequent tissue degeneration [29]. Recent lipidomics studies associate the SARS-CoV-2 virus with profound alterations in lipid metabolism in COVID-19 patients compared to healthy controls [30,31,32,33]. Table 4 lists the characteristic LPC and PC peaks observed in COVID-19 patients and non-COVID-19 subjects, which were detected in positive-ion matrix-assisted laser desorption/ionization (MALDI) spectra using 2,5-Dihydroxybenzoic acid (DHB). The most intense peaks were PC 16:0, 18:2 [M + H]^+^ (*m*/*z* 758.6) and PC 16:0, 18:2 [M + Na]^+^ (*m*/*z* 780.6); PC 18:0, 18:2 [M + H]^+^ (*m*/*z* 786.6) and PC 18:0, 18:2 [M + Na]^+^ (*m*/*z* 808.6); LPC 16:0 [M + H]^+^ (*m*/*z* 496.5) and [M + Na]^+^ (*m*/*z* 518.5); and LPC 18:0 [M + H]^+^ (*m*/*z* 524.5) and [M + Na]^+^ (*m*/*z* 546.3). Since other detected PC and LPC peaks had lower intensities, they were excluded from further analysis. The representative spectrum of characteristic PC and LPC peaks from healthy controls is shown in Figure 2A, whereas the spectrum from COVID-19 patients is shown in Figure 2B. As shown in Table 3, COVID-19 patients had a lower PC/LPC ratio compared with healthy subjects (*p* < 0.001).

Correlation analysis revealed a significant positive correlation (*p* < 0.05) between HO-1 and MDA levels (Table 5) and between HO-1 and the PC/LPC ratio, whereas there was no significant correlation between HO-1 and the PAB, AOPPs, or HNE. A significant negative correlation (*p* < 0.05) was observed between HO-1 and GSH/GSSG ratio. Serum levels of HMGB1 were not significantly associated with any of the oxidative stress parameters in any moderate COVID-19 patients. Nevertheless, we observed a negative but insignificant correlation of HMGB1 with PC/LPC and a small, positive correlation with MDA and HNE levels.

In addition to investigating the PC/LPC ratio as a lipid biomarker of inflammation, we also examined the correlation of serum levels of HO-1 and HMGB-1 with levels of Hp and ferritin, the two acute-phase proteins whose serum levels are elevated in the setting of ongoing inflammation. We found that of all patients with moderate COVID-19, 72.7% had elevated Hp levels and 73.9% had elevated ferritin levels (Table 6). Correlation analysis revealed a significant positive correlation (*p* < 0.01) between HO-1 and ferritin and an insignificant, small, and negative correlation between HMGB1 and ferritin. As for the correlation with Hp, both HO-1 and HMGB1 showed a weak, statistically non-significant positive correlation.

Since increased Hp also reflects the presence of oxidative stress [34,35], a correlation analysis was performed to investigate the relationship between Hp and oxidative stress parameters. We found a significant positive correlation of Hp with PAB (r = 0.243) and HNE (r = 0.250) (Figure 3A,B), a small positive correlation with MDA (r = 0.108), and a small negative correlation (r = −0.133) with GSH/GSSG.

## 3. Discussion

The early pathogenesis of SARS-CoV-2 infection activates various immune cells and leads to the increased production of proinflammatory chemokines and cytokines, the excessive production of which can lead to a “cytokine storm.” This uncontrolled systemic inflammatory response causes immunopathological events, leading to severe pneumonia, ARDS, multiple organ failure, and ultimately the death of a patient with COVID-19 [36,37]. To avoid such a scenario, it is necessary to monitor the expression of proinflammatory mediators from the onset of infection and, accordingly, to search for alternative treatment strategies during the period when the human body still has enough time to regulate its immunity to fight the infection [38].

To date, several clinical studies have shown that excessive HMGB1 levels in the serum of patients with COVID-19 positively correlate with increased levels of IL-6, TNF-ɑ, and IL-1, cytokine storms, and disease severity, making HMGB1 an important mediator for the surveillance and regulation of COVID-19 [16,17,39]. This is also supported by recent findings that HMGB1 regulates the expression of angiotensin-converting enzyme 2 (ACE2), the host receptor for SARS-CoV-2, in addition to the expression of proinflammatory cytokines, thus contributing to SARS-CoV-2 infection [40].

In this study, we have shown that in the early stages of SARS-CoV-2 infection there is a significant increase in serum HMGB1 levels compared with uninfected subjects, suggesting a pathological link between HMGB1 and COVID-19. The fact that all of these COVID-19 subjects survived without experiencing severe complications speaks to the HMGB1 levels most likely reflecting the state of the human body in which it is possible to regulate the immune response with standard medical treatments. Considering that the accurate measurement of HMGB1 in human plasma/serum may be a useful strategy for a more effective treatment of COVID-19 disease [41], several papers have indicated the measured HMGB1 levels associated with disease severity. For example, the mean HMGB1 value of 487.6 pg/mL (315.5–721.8) measured in our study correlated with the value determined by Boley et al. [16] in COVID-19 patients with moderate symptoms (514.7 ± 248.9 pg/mL). These HMGB1 values are significantly lower than the 933.2 pg/mL detected in critically ill patients by Sivakorn et al. [14] or 125.4 ng/mL, which were established as cut-off values to distinguish patients at higher risk of death [42]. According to Vicentino et al. [42], measuring serum HMGB1 levels up to day 12 after hospital admission can help guide pharmacological and medical interventions in early phases of COVID-19 recovery with ongoing inflammation or immune dysfunction.

In recent years, studies have shown that the extracellular presence of HO-1 in plasma, urine, and cerebrospinal fluid is frequently associated with the severity, progression, and prognosis of various diseases, indicating its potential as a biological marker of disease and a biomarker of intracellular HO-1 gene activity [43,44]. HO-1 is mainly released into plasma by leukocytes, macrophages, smooth muscle cells, and endothelial cells activated by oxidative stress or inflammation [45]. In this study, changes in serum levels of HO-1 were investigated because it plays a role in heme catabolism and was found to increase the release of proinflammatory heme after hemolysis and further enhance inflammation by hemolytic products in COVID-19 patients, especially in patients with ARDS [46,47]. Based on the clinical data, it is suggested that hemolysis may be one of the aggravating factors for the transition from mild to severe COVID-19 [47]. Hemolysis is evidenced by a slight increase in serum heme, sufficient to increase the expression of HO-1, which could counteract heme and inhibit exacerbated inflammation [48]. One possible explanation for the upregulation of HO-1 is that it occurs due to the oxidative and inflammatory milieu in the alveolar space of COVID-19 patients, which could include the presence of heme [46]. Although the induction of HO-1 and subsequent production of CO, bilirubin, and iron might have beneficial effects on inhibiting cytokine secretion, their overexpression could be harmful. An excessive release of CO may reduce or inhibit the anti-inflammatory response by activating prostaglandin-endoperoxide synthase, thereby increasing the production of proinflammatory cytokines, or by suppressing the hypothalamic–pituitary–adrenal axis [49]. Moreover, a high serum bilirubin concentration could cause irreversible damage to the central nervous system [50]. However, experimental studies have shown that hemin (synthetic heme), as an inducer of HO-1, attenuates cytokine storms and lung injury in animal models of sepsis and renal ischemia–reperfusion injury, suggesting its possible protective role against cytokine storm syndrome caused by COVID-19 [51]. This is supported by the results of cell culture studies in which hemin, which promotes HO-1 overexpression, effectively suppresses SARS-CoV-2 replication by iron and biliverdin [52]. According to Singh et al. [53], the induction of HO-1 may provide protection in the early stages of COVID-19 by inhibiting viral replication through the upregulation of type 1 interferon expression and in later stages by inhibiting inflammation and coagulation. In addition, the activation of HO-1 through the action of bilirubin, whose increased levels may reduce the consequences and inflammation associated with oxidative stress, may have a beneficial effect on preventing the progression of COVID-19 disease [50].

The results of this study showed a significantly higher serum level HO-1 in patients with mild to moderate COVID-19 at hospital admission compared with subjects without COVID-19. This increase was accompanied by significantly elevated serum ferritin levels, but not bilirubin and iron, suggesting that the observed increase in HO-1 is not necessarily associated with increased heme catabolism and hemolysis, which is in contrast to reports in critically ill patients [22]. According to De Martina et al. [54], the pathogenesis of COVID-19 is not necessarily associated with systemic alterations in heme metabolism or function, as previously suggested [55], or at least not in the early stages of COVID-19 disease. In this case, our results suggest that the observed increased concentration of HO-1 is likely part of the inflammatory response. This conclusion is supported by a significant positive correlation of HO-1 with ferritin and to a lesser extent with Hp, the acute phase proteins and inflammatory markers that play a role in modulating the immune response by triggering anti-inflammatory responses and limiting free radical damage [56,57]. Moreover, we found a significant positive correlation between HO-1 and HMGB1. In the pulmonary system, the binding of extracellular HMGB1 to its receptor for advanced glycation end products (RAGE) activates the nuclear factor kappa light chain enhancer of activated B cells (NF-κB) and mitogen-activated protein kinase (MAPK) signaling pathways, leading to the upregulation of HMGB1 and other proinflammatory mediators and promoting the development of ARDS. Recent evidence has shown that there is a positive feedback loop between peroxisome proliferator-activated receptor gamma (PPARγ) and HO-1, enhancing the protective role of PPARγ in inhibiting HMGB1 production and release. Consequently, it was hypothesized that the upregulation of HO-1 through the activation of PPARγ inhibits HMGB1-RAGE signaling and ameliorates the development of ARDS [58]. In this study, we observed that in some cases where higher HO-1 levels were detected HMGB1 increase levels were lower (data not highlighted). This observation remains to be confirmed, but it suggests a potentially important HO-1/HMGB1 relationship in pathological conditions and implies that HO-1 may be a therapeutic target for reducing COVID-19.

Further evidence supports the contention that the increased HO-1 expression in our group of COVID-19 subjects is part of the inflammatory response. Indeed, a change in redox homeostasis in infected cells is one of the key events associated with SARS-CoV-2 infection and is associated with inflammation [59]. The positive correlation of Hp with PAB and HNE observed in this study is consistent with this conclusion. In addition, we found that the disturbed redox balance and increased pro-oxidant state were characterized by a positive correlation of HO-1 with the PC/LPC ratio, a lipid biomarker of inflammation. Phospholipids are essential not only for the formation of cell membranes but also for the regulation of various biological processes, including signal transduction and inflammation [60]. The changes in lipid ratios could indicate fluctuations in related lipid metabolic pathways [61], while the PC/LPC ratio in serum is recognized as an effective biomarker for various diseases [62]. PLA2 is a lipolytic enzyme that regulates the turnover of PC to LPC by removing the acyl group from the sn-2 position on PC, triggering the loss of important membrane phospholipids and the formation of free fatty acids and LPCs [62]. These species and their metabolites interact with other biomolecules and enhance the accumulation of lipid peroxides, such as MDA and HNE, disrupt membrane permeability and ion homeostasis, promote the formation of several cytokines, including IL-8, IL-6, and I -1β, in epithelial and monocyte cells [30,62], and thus contribute to the inflammatory storm observed in COVID-19 patients. Thus, the observed increase in HO-1 in the early phase of COVID-19 disease could reduce inflammation and provide a survival benefit by offering protection from oxidative stress and inflammation or by acting as a chaperone to remove degraded proteins, as demonstrated in many in vitro and in vivo models of inflammation and lung injury [53,63].

In our study, despite a marked increase in HMGB1 level at moderate COVID-19 patients, no significant correlation was found between HMGB1 and other parameters studied, except with HO-1. In the SARS-CoV2 pandemic, HMGB1 was identified as a potential prognostic biomarker and potential target because it reflects the degree of tissue inflammation and can block inflammatory pathways [12]. In patients with severe COVID-19, multiorgan damage occurs due to the massive induction of various types of cell death (necrosis, apoptosis, iron-dependent lipid peroxidation-induced ferroptosis, pyroptosis) associated with the large release of HMGB1, which is responsible for the development of epithelial barrier failure, organ dysfunction, and even death. HMGB1, released in small amounts during infection, can mediate disease by activating innate immunity through activities that contribute to inflammatory responses that can be remediated [64]. Therefore, the detected HMGB1 levels in mild to moderate COVID-19 patients at hospital admission most likely reflect the activity of the innate immune system and represent a framework of levels within which disease can be controlled.

## 4. Materials and Methods

### 4.1. Patient Selection

This retrospective cross-sectional study included 160 randomly selected adult patients with laboratory-confirmed SARS-CoV-2 infection and mild (N = 96) to moderate (N = 64) COVID-19 disease hospitalized at the Clinic for Pulmonary Diseases, University Clinical Center of Serbia (Belgrade), from November 2020 to January 2021. The severity of COVID-19 disease at the time of the patient’s admission to the hospital was determined based on a chest CT, the Pneumonia Severity Index—PSI [65], and the CURB 65 score [66] according to the NIH guidelines for COVID-19 [2] and the World Health Organization interim guidelines (WHO) [67]. Patients with mild illness exhibit mild clinical symptoms (e.g., fever, cough, sore throat, malaise, headache, muscle pain, nausea, vomiting, diarrhea, loss of taste and smell). They do not have shortness of breath, dyspnea on exertion, or abnormal lung CT imaging. Moderate illness is defined as evidence of lower respiratory disease during clinical assessment or imaging, with SpO2 ≥ 94% on room air [2]. The time from symptom onset to hospital admission ranged from 4 to 9 days. A peripheral blood sample was obtained on admission to the hospital, and laboratory data were analyzed immediately. Data on patients, including basic demographics (sex, age), laboratory results (Hp, iron (Fe), ferritin, total bilirubin (TB)), and comorbidities, were retrospectively collected from electronic health records. The mean age of patients was 62 years (20–83) for female patients (37.5%) and 62 years (19–91) for male patients (62.5%). The laboratory results were interpreted using reference values. Patients had no history of malignancy, coagulopathy, intestinal disease, severe metabolic syndrome, uncontrolled hypertension, uncontrolled diabetes, chronic liver failure, COPD, pulmonary fibrosis, decompensated congestive heart disease, CKD, or psychosis. Sera from healthy subjects (N = 30) were obtained from the Blood Transfusion Institute of Serbia. Cases and controls were matched for sex and age. For sex, exact matches were used. For age, exact matches were preferred, and when this was not possible, approximate matches were used. 

### 4.2. Determination of HMGB1 and HO-1 Concentrations in Serum

Serum HMGB-1 and HO-1 concentrations were determined via an enzyme-linked immunosorbent assay (ELISA) using the Human Heme Oxygenase 1 (HO-1) SimpleStep ELISA^®^ Kit (ab207621, Abcam, Cambridge, UK, sensitivity 7.9 pg/mL) and the Human HMGB1 ELISA Kit (EH0884, FineTest, Wuhan Fine Biotech Co., Ltd., Wuhan, China, sensitivity < 18.75 pg/mL) according to the manufacturer’s protocol. Optical densities were measured at 450 nm using a Sunrice RC Elisa Microplate Reader (Tecan, Austria HmbH, Grödig, Austria). ELISA data were analyzed using GainData (Arigo Biolaboratories, Hsinchu, Taiwan, https://www.arigobio.com/elisa-analysis, accessed on 3 December 2022).

### 4.3. Determination of Biomarkers for Oxidative Stress

The prooxidant/antioxidant balance (PAB) was determined according to the methodology described by Alamdari et al. [68]. The working solution, consisting of a 3,3′,5,5′-tetramethylbenzidine (TMB) cation solution and TMB solution, was combined with the sample, standard, or blank and incubated in the dark at 37 °C for 12 min. The reaction was stopped by adding 2 N HCl, and the absorbance was measured at 450 nm on a microplate reader (WALLAC 1420-Victor2 Multilabel Counter, PerkinElmer, Waltham, MA, USA). PAB values were then calculated and expressed in arbitrary units (HK).

Advanced oxidation protein product (AOPP) levels were determined using the Witko-Sarsat method [69]. Serum samples were first diluted in a phosphate buffer and then added to a chloramine T solution. After an incubation period, the reaction with CH_3_COOH was stopped and the absorbance was measured at 340 nm on a microplate reader (WALLAC 1420-Victor2 Multilabel Counter, PerkinElmer, USA). AOPP concentrations were expressed as μmol/L chloramine T equivalents.

The levels of GSH and GSSG were determined according to the method described by Salbitani et al. [70]. Samples were mixed with a Na-phosphate buffer containing ethylenediaminetetraacetic acid (EDTA) and 5,50-dithio-bis-2-nitrobenzoic acid (DTNB), and the GSH concentration was measured after 5 min at 412 nm using a microplate reader (WALLAC1420-Victor2 Multilabel Counter, PerkinElmer, USA). To quantify the total amount of glutathione (GSH plus GSSG), NADPH and GR were then added to the reaction mixture. After 30 min incubation, the absorbance was measured at 412 nm, and the concentrations of GSH and GSSG were calculated and expressed in µM.

The modified method of Gerard-Monnier et al. [71] was used to determine the concentrations of MDA and HNE. A mixture of 1-methyl-2-phenylindole and acetonitrile/methanol was added to the sample, followed by the addition of HCl to start the MDA assay or methane sulfonic acid and FeCl_3_ for the HNE assay. The reaction mixture was incubated at 45 °C for 40 min, after which the absorbance was measured at 586 nm using a microplate reader (WALLAC 1420-Victor2 Multilabel Counter, PerkinElmer, USA). MDA/HNE concentrations were then calculated using the corresponding standard curves and expressed in µM.

### 4.4. Determination of the PC/LPC Ratio in Serum via MALDI-TOF Mass Spectrometry

The Folch procedure [72] was used to prepare total lipid extracts from serum. The dried organic extracts were resuspended in a 0.5 M DHB matrix solution, and 1 μL of the obtained samples was added to the wells of the stainless-steel target plate [73]. Mass spectra were acquired using a MALDI-TOF (matrix assisted laser desorption/ionisation time of flight) Voyager-DE PRO mass spectrometer (AB-Sciex, Framingham, MA, USA), while the spectra processing of phosphatidylcholine/lysophosphatydilcholine (PC/LPC) was performed using Data Explorer software version 4.9 (Applied Biosystems, Waltham, MA, USA).

### 4.5. Statistical Evaluation

The results are presented as count (%), mean ± standard deviation, or median (25th-75th Percentile) depending on the data type and distribution. The groups are compared using parametric (*t* test) and nonparametric (Mann–Whitney U test) tests. To assess the correlation between variables, parametric Pearson and nonparametric Spearman correlation were used. The type of test depended on the data type and distribution. Numeric variables were evaluated using descriptive statistics, histograms, boxplots, and QQ plots to assess the normality of distribution. In cases where distribution was not normal, alternative nonparametric tests were used. To stabilize variances, some variables were transformed using logarithmic transformation. All *p* values less than 0.05 were considered significant. All data were analyzed using SPSS 29.0 (IBM Corp. Released 2023. IBM SPSS Statistics for Windows, Version 20.0. Armonk, NY, USA: IBM Corp.) and R 3.4.2. (R Core Team (2017). R: A language and environment for statistical computing. R Foundation for Statistical Computing, Vienna, Austria. URL https://www.R-project.org/).

## 5. Conclusions

Our study suggests that HMGB1 and HO-1 in serum could be useful inflammatory indicators in the early stage of COVID-19 infection and could be included in the classification criteria of COVID-19 patients on hospital admission. An original finding of our study was the association of HO-1 with HMGB1 levels, suggesting that targeting the HO-1/HMGB1 relationship may provide a new line of support against the transition from mild/moderate to severe COVID-19 development. Future studies should focus on determining the practical clinical value of these proteins in different clinical settings and developing a scoring system with other serum biomarkers.

## Figures and Tables

**Figure 1 ijms-24-13164-f001:**
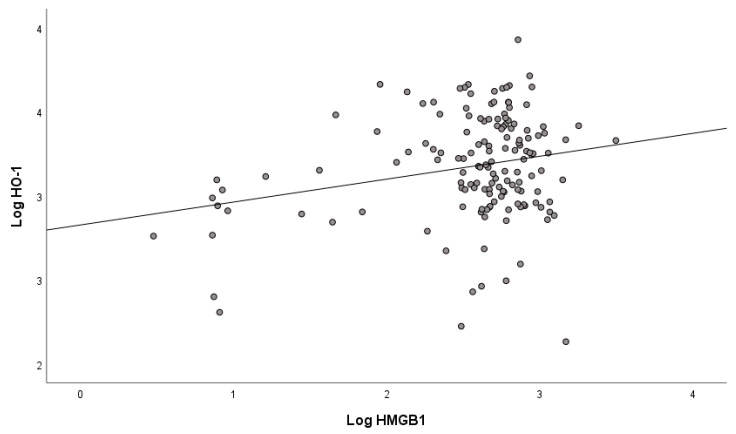
Correlation between HMGB1 and HO-1 in COVID-19 patients at hospital admission.

**Figure 2 ijms-24-13164-f002:**
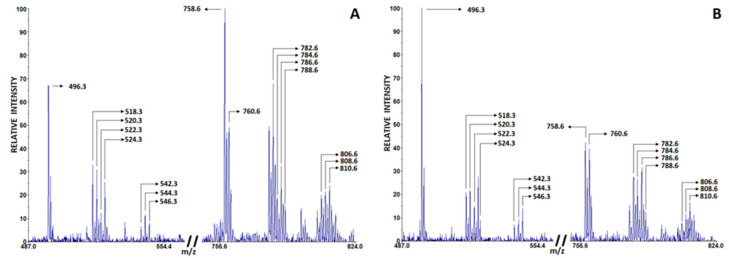
Characteristic phosphatidylcholine (PC) and lysophosphatidylcholine (LPC) peaks in serum of healthy controls (**A**) and COVID-19 patients (**B**) detected in positive mode using DHB matrix.

**Figure 3 ijms-24-13164-f003:**
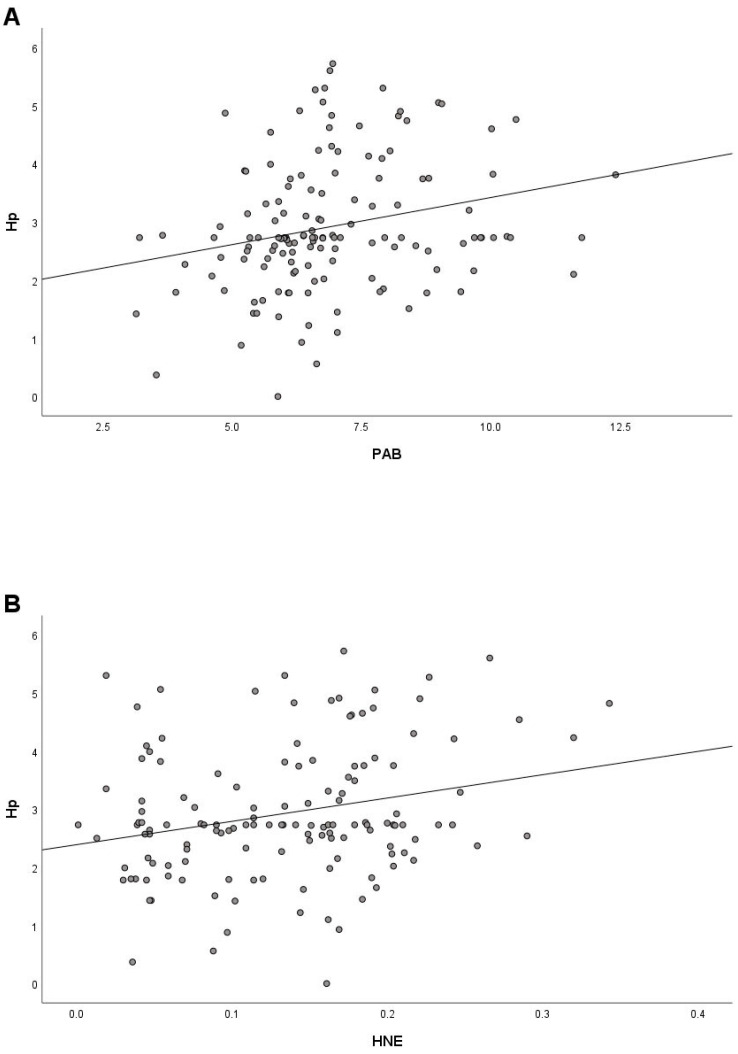
Correlation between Hp and PAB (**A**) and Hp and HNE (**B**) in COVID-19 patients at hospital admission.

**Table 1 ijms-24-13164-t001:** Serum concentrations of HMGB1 and HO-1 in COVID-19 patients at hospital admission.

	COVID-19	Non-COVID-19	*p* Value
HMGB1 (pg/mL)	487.6 (315.5–721.8)	43.1 (18.6–159.9)	<0.001 ^a^
HO-1 (pg/mL)	1497.7 (922.2–2575.3)	756.1 (163.2–1315.8)	0.002 ^a^

^a^ Mann–Whitney U test; descriptive statistics presented as median (25–75th percentile).

**Table 2 ijms-24-13164-t002:** Gender-dependent differences in serum concentrations of HMGB1 and HO-1 in COVID-19 patients.

	Males (n = 100)	Females (n = 60)	*p* Value
HMGB1 (pg/mL)	470.0 (307.2–721.8)	519.3 (340.8–735.5)	0.362 ^a^
HO-1 (pg/mL)	1820.5 (1113.8–2673.2)	1121.7 (854.5–1843.6)	0.005 ^a^

^a^ Mann–Whitney U test; descriptive statistics presented as median (25–75th percentile).

**Table 3 ijms-24-13164-t003:** Oxidative stress parameter values in COVID-19 patients at hospital admission.

Oxidative Stress Markers	COVID-19	Non-COVID-19	*p* Value
PAB (HKU)	6.94 ± 1.75	5.52 ± 1.56	<0.001 ^b^
AOPP (μmol/L)	4.88 ± 1.61	3.45 ± 0.91	<0.001 ^b^
GSH/GSSG	1.09 ± 1.47	1.65 ± 0.58	0.059 ^b^
MDA (μmol/L)	0.137 (0.117–0.180)	0.118 (0.114–0.125)	0.002 ^a^
HNE (µg/mL)	0.140 (0.072–0.181)	0.023 (0.017–0.026)	<0.001 ^a^
PC/LPC	0.667 ± 0.195	1.151 ± 0.129	<0.001 ^b^

^a^ Mann–Whitney U test; ^b^ independent samples *t* test; descriptive statistics presented as mean ± sd or as median (25–75th percentile).

**Table 4 ijms-24-13164-t004:** List of characteristic phosphatidylcholine (PC) and lysophosphatidylcholine (LPC) species identified by MALDI TOF MS detected in positive-ion mode.

Phospholipid Class	Adduct	Signal Position (*m*/*z*)
LPC (16:0)	H^+^	496.3
LPC (16:0)	Na^+^	518.3
LPC (18:2)	H^+^	520.3
LPC (18:1)	H^+^	522.3
LPC (18:0)	H^+^	524.3
LPC (18:2)	Na^+^	542.3
LPC (18:1)	Na^+^	544.3
LPC (18:0)	Na^+^	546.3
PC (34:2)	H^+^	758.6
PC (34:1)	H^+^	760.6
PC (34:1)	Na^+^	782.6
PC (36:3)	H^+^	784.6
PC (36:2)	H^+^	786.6
PC (36:1)	H^+^	788.6
PC (36:3)	Na^+^	806.6
PC (36:2)	Na^+^	808.6
PC (38:4)	H^+^	810.6

**Table 5 ijms-24-13164-t005:** Correlation between oxidative stress parameters in COVID-19 patients with HO-1 and HMGB1.

	HO-1	Log HO-1	HMGB1	Log HMGB1
PAB ^p^	0.004	0.034	−0.078	0.021
AOPP ^p^	−0.42	0.047	0.097	0.087
GSH/GSSG ^p^	−0.077	−0.167 *	0.028	−0.035
MDA ^s^	0.183 *	0.183 *	0.050	0.050
Log MDA ^p^	0.092	0.184 *	0.065	0.148
HNE ^p^	0.014	0.079	0.103	0.137
PC/LPC ^p^	0.129	0.194 *	−0.142	−0.025

^p^ Pearson correlation coefficient; ^s^ Spearman correlation coefficient; * correlation is significant at the 0.05 level (*p* < 0.05).

**Table 6 ijms-24-13164-t006:** Laboratory values of serum haptoglobin (Hp), free iron (Fe), ferritin, and total bilirubin (TB) levels in COVID-19 patients and their correlations with HMGB1 and HO-1.

	Descriptive Stat.	Elevated (%)	HO-1	Log HO-1	HMGB1	Log HMGB1
Hp ^p^	2.92 ± 1.13	117 (72.7%)	0.101	0.045	0.112	0.121
Fe ^p^	15.47 ± 7.82	6 (3.7%)	0.063	0.021	0.036	−0.038
Ferritin ^s^	527 (274–968)	119 (73.9%)	0.297 *	0.263 *	−0.032	−0.116
Log ferritin^p^			0.297 *	0.251 *	−0.032	−0.086
TB ^s^	9.9 (7.2–13.8)	11 (6.8%)	0.082	0.145	0.084	0.021
Log TB ^p^			0.082	0.092	0.084	0.033

^p^ Pearson correlation coefficient; ^s^ Spearman correlation coefficient; * correlation is significant at the 0.05 level (*p* < 0.05).

## Data Availability

Enquiries about data availability should be directed to the authors. The data are not publicly available due to privacy and ethical restrictions.

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
