# Peer review of "Serum High-Mobility Group Box 1 and Heme Oxygenase-1 as Biomarkers in COVID-19 Patients at Hospital Admission"

_ijms, 2023, doi:10.3390/ijms241713164_

Round 1
Reviewer 1 Report
Ilijanana et al. conducted a study to monitor serum levels of several markers in early-stage of COVID-19 patients mild/moderate symptoms. They compared these levels with non-COVID-19 controls and performed correlation studies. The authors discovered higher levels of HMGB1 and HO-1 in the early phase of mild/severe COVID-19 subjects compared to healthy controls. They also observed a positive correlation between HMGB1 and HO-1. This study suggests that these biomarkers could be candidates for predicting inflammation severity in the early phase of COVID-19 and potentially included in the classification criteria for hospitalized COVID-19 patients.
However, there are several concerns regarding the study's design, analysis, presentation, and interpretation:
Patient selection: The study included 160 adult patients with laboratory-confirmed SARS-CoV-2 infection and moderate-severe COVID-19 disease in the method. The author inconsistently stated those 160 subjects are mild-moderate patients in the manuscript. It is unclear how many patients were classified as mild or moderate or severe. The manuscript lacks information about the control group, such as whether it was matched for age and gender with the COVID-19 group.
Severity comparison: Since the authors aim to predict COVID-19 severity using biomarkers, it would be beneficial to compare biomarker levels among groups with different severity levels. Alternatively, if multiple blood samples were collected from subjects at different time points, assessing the dynamic changes of these markers could provide valuable insights.
Statistical analysis explanation: The authors should provide details in the methods section about the criteria used to choose specific statistical analyses for comparisons, such as the rationale behind using Mann-Whitney U test, independent sample t-test, Pearson correlation, or Spearman correlation.
Previous reports and alternative analyses: While the significant increase in HMGB1 and HO-1 levels in COVID-19 subjects is interesting, it has been reported by other studies in severe and critical cases. Moreover, to explore biomarkers as predictors in COVID-19, several alternative analyses would be more appropriate to identify their significance and predictive value. Examples include univariate and multivariate analysis to assess associations with specific outcomes, Receiver Operating Characteristic (ROC) analysis and Area under the Curve (AUC) analysis, and machine learning techniques.
Presentation of correlation analysis: To improve clarity, it would be helpful to include the r and p values in the tables for the correlation analysis.
Overall, addressing these concerns and considering alternative analyses would enhance the study's design and provide stronger support for the significance and predictive value of the examined biomarkers in COVID-19.
I didn't notice grammar errors in the manuscript but it would be good to revise the manuscript using succinct and concise language.
Author Response
List of Responses to Rewiever 1
- Patient selection: The study included 160 adult patients with laboratory-confirmed SARS-CoV-2 infection and moderate-severe COVID-19 disease in the method. The author inconsistently stated those 160 subjects are mild-moderate patients in the manuscript. It is unclear how many patients were classified as mild or moderate or severe. The manuscript lacks information about the control group, such as whether it was matched for age and gender with the COVID-19 group.
Our response:
- a) As we have already said in the paper and you also pointed out, a significant increase in HMGB1 and HO-1 levels in people with COVID-19 has been reported in other studies, mostly in severe and critical cases. Therefore, the basis of this work was the idea to determine the values of HMGB1 and HO1 in the serum of patients with only mild to moderate symptoms of COVID-19, in one time point, i.e. upon admission to the hospital, who survived without additional complications during their hospital stay. We wanted to determine levels of HMGB1 and HO1, which maintenance with clinical treatments could potentially keep the disease under control. Randomly, 96 mild and 64 moderate patients were selected, in whom the parameters from the work were determined separately (Table 1). Patients with moderate COVID-19 have significantly higher mean ferritin and significantly lower mean AOPP and mean MDA and HNE. HO-1 and GSH/GSSG levels are also higher in moderate patients, but statistical significance in these analyzes is not less than 0.05. As there is no difference in the levels of HMGB1 and HO1 between mild and moderate COVID-19 patients, which was confirmed by logistic regression analysis (table 2), we studied mild to moderate patients as one group and observed the results only in the context of patients with COVID-19 and non-Covid-19. There are no severe or critical patients in our study. If you agree, we would add these two tables to the supplement materials.
- b) In the materials and methods subsection- patient selection, we have added information related to number of mild and moderate patients which were included in COVID-19 group and information related to age and gender matching (marked in red).
.
Table 1. Examined biomarkers level in mild and moderate COVID-19 patients
|
|
Mild (n=96) |
Moderate (n=64) |
p value |
|
HMGB1 (pg/ml) |
496.1 (344.3-739.8) |
470.3 (184.6-634.3) |
0.202 a |
|
HO-1 (pg/ml) |
1376.3 (908.9-2152.0) |
1794.2 (1059.4-3063.9) |
0.063 a |
|
Hp |
2.91±1.01 |
2.94±1.28 |
0.873 b |
|
Fe |
15.64±7.08 |
15.21±8.89 |
0.744 b |
|
Ferritin |
395 (194.5-751.6) |
840.1 (424.6-1675.6) |
<0.001 a |
|
TB |
9.95 (7.10-12.80) |
9.8 (7.5-15.5) |
0.346 a |
|
PAB (HKU) |
6.89±1.49 |
6.97±1.98 |
0.782 b |
|
AOPP (μmol/l) |
5.11±1.57 |
4.38±1.41 |
0.003 b |
|
MDA (μmol/l) |
0.139 (0.120-0.181) |
0.129 (0.110-0.165) |
0.018 a |
|
HNE (µg/ml) |
0.145 (0.090-0.184) |
0.114 (0.046-0.172) |
0.044 a |
|
GSH/GSSG |
0.855±0.268 |
0.948±0.331 |
0.073 b |
|
PC/LPC |
0.671±0.173 |
0.658±0.219 |
0.696 b |
aMann-Whitney U test; bt test; descriptive statistics presented as median (25-75th percentile) or mean±sd
Table 2. The logistic regression analysis in mild and moderate COVID-19 patients
|
MODEL |
Enter |
|
Backward |
|
|
OR (95% CI) |
p value |
OR (95% CI) |
p value |
|
|
Log Ferritin |
3.596 (1.359-9.514) |
0.010 |
4.437 (1.724-11.424) |
0.002 |
|
Log HO-1 |
2.123 (0.594-7.591) |
0.247 |
|
|
|
AOPP |
0.865 (0.635-1.179) |
0.359 |
|
|
|
HNE |
0.010 (0.001-3.848) |
0.129 |
|
|
|
Log MDA |
0.100 (0.003-3.534) |
0.206 |
0.004 (0.001-0.975) |
0.049 |
|
GSH/GSSG |
1.920 (0.450-8.198) |
0.378 |
0.040 (0.002-0.767) |
0.033 |
aBackward method was performed with 0.10 sigificance as a cut off for variable elimination
- Severity comparison: Since the authors aim to predict COVID-19 severity using biomarkers, it would be beneficial to compare biomarker levels among groups with different severity levels. Alternatively, if multiple blood samples were collected from subjects at different time points, assessing the dynamic changes of these markers could provide valuable insights.
Our response: This study was not prospective. As we state in our previous response, we studied mild to moderate patients as one group, in only one time point. There are no groups with different severity levels.
- Statistical analysis explanation: The authors should provide details in the methods section about the criteria used to choose specific statistical analyses for comparisons, such as the rationale behind using Mann-Whitney U test, independent sample t-test, Pearson correlation, or Spearman correlation.
Our response: In the subsection Statistical explanation, in the methods section, we tried to give a better explanation for statistical analysis of the obtained data. We have added the following:
“Results are presented as count (%), mean ± standard deviation or median (25th-75th percentile) depending on data type and distribution. Groups are compared using parametric (t test) and nonparametric (Mann-Whitney U test) tests. To assess correlation between variables parametric Pearson and non-parametric Spearman correlation was used. The type of test depended on data type and distribution. Numeric variables were evaluated using descriptive statistics, histogram, boxplot and QQ plot to assess the normality of distribution. In cases where distribution was not normal, the alternative non-parametric tests were used. In order to stabilize variances, some variables were transformed using logarithmic transformation. All p values less than 0.05 were considered significant. All data were analyzed using SPSS 29.0 (IBM Corp. Released 2023. IBM SPSS Statistics for Windows, Version 20.0. Armonk, NY: IBM Corp.) and R 3.4.2. (R Core Team (2017). R: A language and environment for statistical computing. R Foundation for Statistical Computing, Vienna, Austria. URL https://www.R-project.org/.).”
- 4. Previous reports and alternative analyses: While the significant increase in HMGB1 and HO-1 levels in COVID-19 subjects is interesting, it has been reported by other studies in severe and critical cases. Moreover, to explore biomarkers as predictors in COVID-19, several alternative analyses would be more appropriate to identify their significance and predictive value. Examples include univariate and multivariate analysis to assess associations with specific outcomes, Receiver Operating Characteristic (ROC) analysis and Area under the Curve (AUC) analysis, and machine learning techniques.
Our response: In the title, we used the term predictive biomarkers, thinking that the obtained values for elevated levels of HMGB1 and HO-1 in the examined COVID-19 cases can potentially predict a positive outcome (survival) of patients during COVID-19 managing. Bearing in mind what we stated in the response to remark 1, we propose correction of the title of the manuscript , abstract and conclusion and removing the term predictive. We hope that you will agree with those changes.
- Presentation of correlation analysis: To improve clarity, it would be helpful to include the r and p values in the tables for the correlation analysis.
Our response: In all correlation tables correlation coefficients are presented with * as mark for significance. In all tables instead of p Pearson correlation; s Spearman correlation, *Correlation is significant at the 0.05 level, the following is added to be more interpretable p Pearson correlation coefficient; s Spearman correlation coefficient, *Correlation is significant at the 0.05 level (p<0.05)
Finally, I wish to express my gratitude to your opinion and constructive suggestions for improving the manuscript.
Sincerely,
Dr. Ilijana Grigorov
Reviewer 2 Report
Please describe the study design in the abstract. (retrospective or prospective, random selection, study time window, etc.)
Abbreviations need to be spelled out at their first appearance. Examples: MDA, PC /LPC, GSH/GSSG)
Line 183, criteria to define moderate COVID-19 patients need to be listed.
HMGB1 and HO-1were shown to correlate with oxidative stress parameters, so in order to demonstrate they have independent and additional predictive value for COVID severity and outcome, a multi-variate model needs to be run to show adjusted association of HMGB1 and HO-1 with COVID.
NA
Author Response
List of Responses to Rewiever 2
1.Please describe the study design in the abstract. (retrospective or prospective, random selection, study time window, etc.)
Our response: According to your suggestion we described our study design in the abstract as retrospective, cross-sectional, with randomly selected COVID-19 patients from one time point, at hospital admission.
- Abbreviations need to be spelled out at their first appearance. Examples: MDA, PC /LPC, GSH/GSSG)
Our response: Adequate changes was made through text (marked in red)
- Line 183, criteria to define moderate COVID-19 patients need to be listed.
Our response: I'm sorry, but is it possible that there was a misunderstanding regarding line 183?
As we mentioned in subsection of materials and methods – Patients selection, the mild and moderate severity of COVID-19 disease at the time of the patient's admission to the hospital was determined based on the Chest CT, the Pneumonia Severity Index- PSI and the CURB 65 score according to the NIH guidelines for COVID-19 and the World Health Organization interim guidelines (WHO). Following text were added (marked as red): “Patients with mild illness exhibit a mild clinical symptoms (e.g., fever, cough, sore throat, malaise, headache, muscle pain, nausea, vomiting, diarrhea, loss of taste and smell). They do not have shortness of breath, dyspnea on exertion, or abnormal lung CT imaging. Moderate illness is defined as evidence of lower respiratory disease during clinical assessment or imaging, with SpO2 ≥94% on room air. “
- HMGB1 and HO-1were shown to correlate with oxidative stress parameters, so in order to demonstrate they have independent and additional predictive value for COVID severity and outcome, a multi-variate model needs to be run to show adjusted association of HMGB1 and HO-1 with COVID.
Our response: Thank you for this comment. The basis of this work was the idea to determine the values of HMGB1 and HO1 in the serum of patients with only mild to moderate symptoms of COVID-19, in one time point, i.e. upon admission to the hospital, who survived without additional complications during their hospital stay. We wanted to determine levels of HMGB1 and HO1, which maintenance with clinical treatments could potentially keep the disease under control. Randomly, 96 mild and 64 moderate patients were selected, in whom the parameters from the work were determined separately (Table 1). Patients with moderate COVID-19 have significantly higher mean ferritin and significantly lower mean AOPP and mean MDA and HNE. HO-1 and GSH/GSSG levels are also higher in moderate patients, but statistical significance in these analyzes is not less than 0.05. As there is no difference in the levels of HMGB1 and HO1 between mild and moderate COVID-19 patients, which was confirmed by logistic regression analysis (table 2), we studied mild to moderate patients as one group and observed the results only in the context of patients with COVID-19 and non-Covid-19. There are no severe or critical ill patients in our study. If you agree, we would add these two tables to the supplement materials.
Table 1. Examined biomarkers level in mild and moderate COVID-19 patients
|
|
Mild (n=96) |
Moderate (n=64) |
p value |
|
HMGB1 (pg/ml) |
496.1 (344.3-739.8) |
470.3 (184.6-634.3) |
0.202 a |
|
HO-1 (pg/ml) |
1376.3 (908.9-2152.0) |
1794.2 (1059.4-3063.9) |
0.063 a |
|
Hp |
2.91±1.01 |
2.94±1.28 |
0.873 b |
|
Fe |
15.64±7.08 |
15.21±8.89 |
0.744 b |
|
Ferritin |
395 (194.5-751.6) |
840.1 (424.6-1675.6) |
<0.001 a |
|
TB |
9.95 (7.10-12.80) |
9.8 (7.5-15.5) |
0.346 a |
|
PAB (HKU) |
6.89±1.49 |
6.97±1.98 |
0.782 b |
|
AOPP (μmol/l) |
5.11±1.57 |
4.38±1.41 |
0.003 b |
|
MDA (μmol/l) |
0.139 (0.120-0.181) |
0.129 (0.110-0.165) |
0.018 a |
|
HNE (µg/ml) |
0.145 (0.090-0.184) |
0.114 (0.046-0.172) |
0.044 a |
|
GSH/GSSG |
0.855±0.268 |
0.948±0.331 |
0.073 b |
|
PC/LPC |
0.671±0.173 |
0.658±0.219 |
0.696 b |
aMann-Whitney U test; bt test; descriptive statistics presented as median (25-75th percentile) or mean±sd
Table 2. The logistic regression analysis in mild and moderate COVID-19 patients
|
MODEL |
Enter |
|
Backward |
|
|
OR (95% CI) |
p value |
OR (95% CI) |
p value |
|
|
Log Ferritin |
3.596 (1.359-9.514) |
0.010 |
4.437 (1.724-11.424) |
0.002 |
|
Log HO-1 |
2.123 (0.594-7.591) |
0.247 |
|
|
|
AOPP |
0.865 (0.635-1.179) |
0.359 |
|
|
|
HNE |
0.010 (0.001-3.848) |
0.129 |
|
|
|
Log MDA |
0.100 (0.003-3.534) |
0.206 |
0.004 (0.001-0.975) |
0.049 |
|
GSH/GSSG |
1.920 (0.450-8.198) |
0.378 |
0.040 (0.002-0.767) |
0.033 |
aBackward method was performed with 0.10 sigificance as a cut off for variable elimination
In the title, we used the term predictive biomarkers, thinking that the obtained values for elevated levels of HMGB1 and HO-1 in the examined COVID-19 cases can potentially predict a positive outcome (survival) of patients during COVID-19 managing. We propose correction of the title of the manuscript , abstract and conclusion and removing the term predictive. We hope that you will agree with those changes.
Finally, I wish to express my gratitude to your opinion and constructive suggestions for improving the manuscript.
Sincerely,
Dr. Ilijana Grigorov
Round 2
Reviewer 1 Report
The paper has undergone significant enhancements.